# Evidence on the links between water insecurity, inadequate sanitation and mental health: A systematic review and meta-analysis

Joan J. Kimutai[1,2]*, Crick Lund[3,4], Wilkister N. Moturi[5], Seble Shewangizaw[2], Merga Feyasa[1], Charlotte Hanlon[1,2,3]

1 Centre for Innovative Drug Development and Therapeutic Trials for Africa (CDT-Africa), College of Health Sciences, Addis Ababa University, Addis Ababa, Ethiopia, 2 Department of Psychiatry, School of Medicine, College of Health Sciences, Addis Ababa University, Addis Ababa, Ethiopia, 3 Centre for Global Mental Health, Health Service and Population Research Department, Institute of Psychiatry, Psychology and Neuroscience, King's College London, London, United Kingdom, 4 Alan J Flisher Centre for Public Mental Health, Department of Psychiatry and Mental Health, University of Cape Town, Cape Town, South Africa, 5 Department of Environmental Science, Faculty of Environment and Resource Development, Egerton University, Njoro, Kenya

* joan.mining@gmail.com

**Data Availability Statement:** All relevant data are within the paper and its  Supporting Information files.

## Abstract

### Background

Water insecurity and inadequate sanitation have adverse impacts on the mental health of individuals.

### Objective

To review and synthesize evidence on the relationship between water insecurity, inadequate sanitation, and mental health globally.

### Data sources

Relevant studies were identified by searching PubMed, PsycINFO, and EMBASE databases from inception up to March 2023.

### Study eligibility criteria

Only quantitative studies were included. The exposure was water insecurity and or inadequate sanitation. The outcome was common mental disorders (CMD: depression or anxiety), mental distress, mental health or well-being. There was no restriction on geographical location.

### Participants

General population or people attending health facilities or other services.

### Exposure

Water insecurity and/ or inadequate sanitation.

**Funding:** The authors received no specific funding for this work.

**Competing interests:** The authors have declared that no competing interests exist.

## Risk of bias

The effective Public Health Practice Project (EPHPP) assessment tool was used to assess quality of selected studies.

## Synthesis of results

A meta-analysis was conducted using a random effects statistical model.

## Results

Twenty-five studies were included, with 23,103 participants from 16 countries in three continents: Africa (Kenya, Ethiopia, Ghana, Uganda, South Africa, Malawi, Mozambique, and Lesotho), Asia (Nepal, Bangladesh, India, and Iran) and the Americas (Brazil, Haiti, Bolivia and Vietnam). There was a statistically significant association between water insecurity and CMD symptoms. Nine studies reported a continuous outcome (5,248 participants): overall standardized mean difference (SMD = 1.38; 95% CI = 0.88, 1.87). Five studies reported a binary outcome (5,776 participants): odds ratio 5.03; 95% CI = 2.26, 11.18. There was a statistically significant association between inadequate sanitation and CMD symptoms (7415 participants), overall SMD = 5.36; 95% CI = 2.51, 8.20.

## Limitations

Most of the included studies were cross-sectional which were unable to examine temporal relationships.

## Conclusions

Water insecurity and inadequate sanitation contribute to poorer mental health globally.

## Implications of key findings

Interventions to provide basic water, sanitation and psychosocial support, could substantially contribute to reducing the burden of CMD alongside other health and social benefits.

## Trial registration

**PROSPERO registration number:** CRD42022322528.

## Introduction

There is increased research and policy focus on understanding the social determinants of mental health to improve intervention efforts [1]. Depression and anxiety (termed 'common mental disorders'; CMD) are the leading contributors to mental ill-health globally, with a prevalence of 3.4% and 3.8%, respectively [2]. The population burden of depression and anxiety is closely associated with social determinants, including poverty, living environment, violence and migration [1].

An estimated 4 billion people in the world (two thirds of the world's population), are affected by water scarcity every year [3] and approximately 663 million people do not have access to safe drinking water [4]. Using microbiologically unsafe drinking water is associated

with increased burden of diarrheal diseases, helminth infections and impaired physical health [5]. Water is justifiably recognized in the United Nations Sustainable Development Goals (goal 6) as a vital determining factor for population health and development. Alongside this, an estimated 2.5 billion people worldwide lack access to basic sanitation, defined as an unshared household sanitation facility that hygienically isolates human excreta from human contact [6]. One billion people lack access to any kind of sanitation facility and practice open defecation [7], which is associated with numerous adverse health impacts. Improved sanitation has significant health benefits. Eliminating exposure to human excreta reduces the risk of diarrhoea, schistosomiasis, trachoma and soil-transmitted helminth which can cause stunting, cognitive impairment or death especially among children under the age of five [5].

There has been increasing interest in the direct and indirect influences of water insecurity on mental health. People may be anxious about getting sufficient water or safe water [8]. Water insecurity can exacerbate frustrations, shame, and gender-based violence, as women most often bear responsibility for ensuring household access to clean water [9–11]. In addition, water insecurity can result in mental distress due to the opportunity costs of time spent fetching or queuing for water, for example, leading to less time available for employment and education [12].

Growing research has shown evidence of mental health risks associated with inadequate sanitation. These include feelings of shame if seen by others, restricting intake of water and food to limit defecation and urination, suppressing need to use sanitation because of a non-conducive social and physical environment, fearing or encountering sexual or physical violence when using open defecation sites or feeling incapable of changing sanitation conditions [13–15]. In addition, there is also disgust at seeing or smelling faeces which can trigger anxiety or shame [16].

A scoping review was published in 2017, with the aim of summarizing evidence linking water insecurity and inadequate sanitation to psychosocial distress [17]. This scoping review was limited to published articles from 1980 to March 2016 and identified 15 studies (8 qualitative, 4 mixed-method and 3 quantitative). Four interrelated groups of mental health stressors were identified as associated with lack of safe water and adequate sanitation (financial stressors, social stressors, physical stressors, and stressors related to perceived inequalities). In the six years since publication of this review, there has been increased attention to this area, so it is now appropriate to conduct a systematic review. To the best of our knowledge no systematic review has yet been conducted on the associations between water security, sanitation and common mental disorders. A systematic review would be useful to synthesise knowledge on this area, and to identify research gaps and potential intervention priorities.

## Objective

The objective of this systematic review and meta-analysis was to synthesise evidence pertaining to the relationship between water insecurity, inadequate sanitation and mental health globally.

## Methods

This systematic review built on the previous scoping review by expanding to include all global regions and conducting a meta-analysis.

### Registration and protocol

This systematic review was conducted and reported according to PRISMA guidelines [18]. A protocol was registered on PROSPERO (CRD42022322528) in 2022. As this is a systematic review and a meta-analysis of published articles, ethical approval was not sought.

## PICO (Population intervention/Exposure comparison outcome)

The PICO criteria framework was used to effectively develop our literature search strategy. The population was the general population or people attending health facilities or other services. The exposure was water insecurity and/ or inadequate sanitation. Comparison was between people in more water secure settings and those with more adequate sanitation or a comparison intervention. The outcome was common mental disorders, mental distress, mental health or wellbeing.

## Inclusion and exclusion criteria

Table 1 sets out the inclusion and exclusion criteria for this review. The mental health conditions under consideration were depression, anxiety, stress, somatoform disorders, and common mental disorders. Psychosis, substance abuse, dementia and epilepsy were excluded because there is less robust evidence to support a causal relationship between water insecurity/ inadequate sanitation and these conditions.

## Search strategy

The following electronic databases were used to search for articles; PubMed, PsycINFO, and Embase from inception until the date of search (March 2023). Search strategies were constructed to include both Medical Subject Headings (MESH terms) and free-text terms (title and abstracts). The full search strategy is presented in the supplementary file (S1 Appendix). The following search terms were used 'Water *insecurity' OR 'water scarcity' OR 'access to

**Table 1. Inclusion and exclusion criteria.**

|  | Included | Excluded |
|---|---|---|
| Publication type | Date: from inception until the date of search<br>All studies done globally.<br>Only peer reviewed journals | Conference abstracts |
| Study design | Only quantitative studies:<br>• cross-sectional surveys<br>• case-control studies<br>• cohort studies<br>• intervention studies | Qualitative studies, case studies, case series |
| Study population | Adult population: 15 years and above, depending on how this was defined in a country.<br>If both children and adults included, only included if the results were stratified. | |
| Exposure | Water insecurity and/or inadequate sanitation | |
| Definition of water insecurity | Insecure and uncertain access to adequate water for a healthy lifestyle [19]. | |
| Definition of inadequate sanitation | Insufficient and uncertain access to a conducive environment that respects and responds to individual sanitation needs (urination, defecation and managing menstruation). | |
| Outcome | Mental health conditions (depression, anxiety, stress, somatoform disorders) or mental distress (psychosocial impact) or common mental disorder symptoms or mental health or wellbeing. Measured by self-report or clinical assessment. | Psychosis, substance abuse, dementia, or epilepsy. |
| Definition of mental health | "A state of well-being in which the individual realizes his or her own abilities, can cope with the normal stresses of life, can work productively and fruitfully, and is able to make a contribution to his or her community" [20]. | |

water' OR 'water access' OR 'water availability' OR 'water supply' OR 'water quant*' OR 'water distribution' OR 'sanitation' OR 'open defecation' OR 'toilet facilit*' OR 'latrine' OR 'poor sanitation' AND 'mental health' OR 'mental disorders' OR 'common mental disorders' OR 'depression' OR 'anxiety' OR 'mental distress' OR 'well-being' OR 'wellbeing'. All records were downloaded into an Endnote library to facilitate removal of duplicates and then exported to Excel.

## Screening

First, title and abstract screening was done independently by two reviewers J.K and S.S (PhD students with knowledge, skills and experience in systematic reviewing). Each reviewer independently screened all identified titles and abstracts. The decision to select a retrieved article for further evaluation was based on the eligibility criteria. Where there was any doubt about whether the article should be included, the full text was obtained to inform the decision. Second, full text screening was carried out by two reviewers working independently, with differences resolved by senior authors (C.H, C.L and W.M). Inter-rater reliability was also tested for the full test screening using Cohen's Kappa statistic. The percentage of agreement was 87.4% and Cohen's k was 0.70, indicating substantial agreement.

## Data collection process and data items

Data were extracted into a codebook by two independent reviewers. The extraction code book included name of the authors and year of publication, description of the participants and settings, study design, sampling, exposure variables, outcome measures (including details of instruments used) and results (effect size for each outcome, p-value, confidence interval and standard deviation).

## Quality assessment

Quality assessment of the selected full text articles was done using the Effective Public Health Practice Project (EPHPP) quality assessment tool for quantitative studies [21] presented in the supplementary files (S2 Appendix). Methodological quality (study design, selection bias, confounding, blinding, data collection, and withdrawal/dropouts) of each individual study included in the systematic review was assessed independently by two trained reviewers.

## Effect measure

Effect measures for both continuous and binary outcomes were used. The meta-analysis summary effect measure for continuous outcomes was an estimate of the mean and standard deviation of a distribution of true effects. For the binary outcome, the number of cases in whom there was an event (CMD symptoms) and the sample sizes of both water insecure and or sanitation insecure and water secure and or sanitation secure groups was used for effect measure.

## Synthesis methods

The study characteristics, methodological description (study setting, study population, study design, sampling method, exposure, outcome and measurement tools), and the main findings of included studies were tabulated. This facilitated examination of the PICO elements across studies. We used excel sheets to prepare data for synthesis, where we recorded data on the mean, standard deviation, number of events, and sample size. Data were organised in a form that could be read by the analysis software. We then proceeded to conduct meta-analysis. Review Manager (RevMan) software was used for the meta-analysis. The review manager

software conducts meta-analysis by calculating the standardized mean difference (the sum of the weighted effect sizes, divided by the sum of weightings) when the outcome of research is continuous, and odds ratio when the outcome of research is binary.

A random effects (Der Simonian and Laird) statistical model for meta-analysis was used. The random effects model was used because it assumes the true effect may vary from study to study due to heterogeneity (differences) among studies. The effect estimates of all studies are assumed to be drawn from a normal distribution, and the pooled estimate is the average or mean effect. The effect sizes in the studies used are also assumed to represent a random sample for all possible effect sizes.

The findings from the meta-analysis were presented as a forest plot. A forest plot is a graphical display of findings of meta-analysis summarizing the data of individual studies. It gives a visual suggestion of the amount of heterogeneity and the estimated common effect. A funnel plot was used to look for evidence of publication bias (the tendency of authors to publish significant results from studies). A funnel plot is a graphical representation of the relation between the study's effect size and its precision (effect estimated from individual studies against sample size). An asymmetry funnel plot indicates a possibility of publication bias. Whereas a symmetrical inverted funnel plot indicates no publication bias in the included studies.

RevMan software was used to investigate heterogeneity in the outcomes between studies using the $I^2$ statistics. Heterogeneity between studies was high even after performing random effect meta-analyses. An exploratory post-hoc sub-group and sensitivity analyses were conducted. We grouped the included studies into World Bank classifications by country income level (low-income country, lower-middle-income country, and upper-middle income country).

## Results

### Study characteristics

Out of 6105 non-duplicate records identified, 25 studies were included in the systematic review for meta-analysis, as shown in the PRISMA flow diagram (Fig 1).

The characteristics of the studies included in the review are described in Table 2. Fifteen studies were conducted in lower middle-income countries, while eight were conducted in low-income countries and two in upper middle-income countries. There were no studies from high-income countries. Eighteen studies used a cross-sectional study design, four were cohort studies and three were intervention study. Most (n = 20) studies examined water insecurity only, with three studies investigating inadequate sanitation only and two studies examining both water insecurity and inadequate sanitation. The outcomes examined included common mental disorder symptoms (depression, anxiety and mental distress) and mental well-being. The most commonly used screening tools for the outcomes were the Self Reporting Questionnaire (SRQ-20) [11, 22–24] and Hopkins Symptom Checklist (HSCL) [25–28]. The methodological description and the main findings of the included studies are shown in the Tables 3 and 4.

### Quality assessment

The quality assessment results are shown in Fig 2. The most identified quality issues were blinding and confounding. Studies failed to indicate whether the outcome assessors and study participants were blinded. They also failed to indicate whether confounders were controlled for in the design or analysis stage. Some studies did not mention the number of people who were missing from the source population, arising from non-response. There was also variation in the extent to which studies used standardised and validated measures.

## PRISMA Flowchart

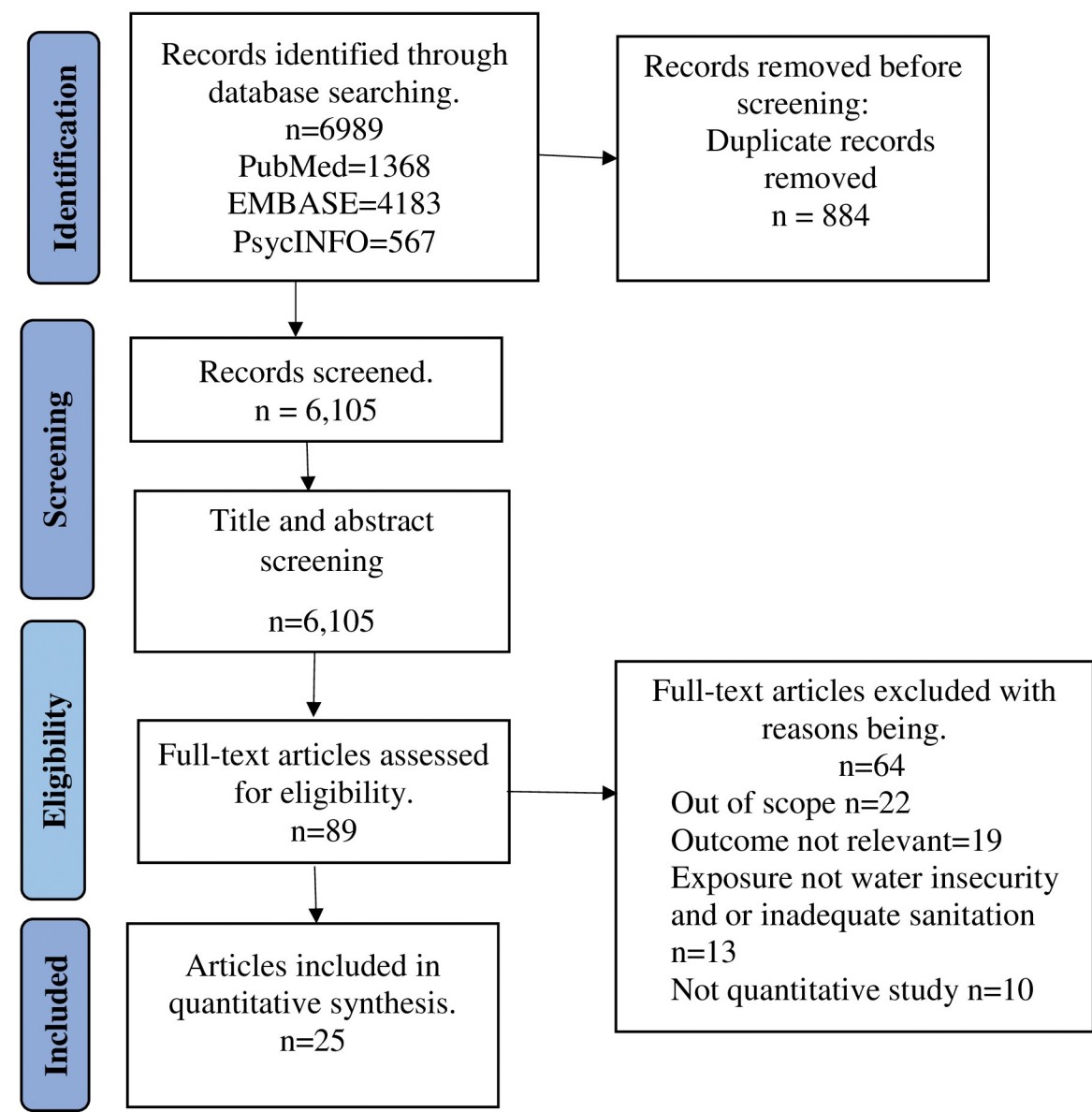

**Fig 1. PRISMA flowchart showing the study selection and screening process.**

### Meta-analysis

**Meta-analysis for continuous outcome.** *Association between water insecurity and common mental disorders (CMD).* Ten studies reported continuous outcomes (Fig 3). The results indicated a statistically significant association between water insecurity and common mental disorder (CMD) symptoms, with a standardized mean difference (SMD) = 1.38 (95% CI = 0.88, 1.87). Heterogeneity between studies was substantial ($I^2$ = 98%).

**Meta-analysis of binary outcomes.** *Association between water insecurity and common mental disorders (CMD).* Six studies reported binary outcomes. The forest plot is shown in

**Table 2. Description of study characteristics.**

| Study characteristic | Number of studies (n = 25) |
|---|---|
| **Africa** | |
| Ethiopia | 5 |
| Ghana | 2 |
| Kenya | 3 |
| Lesotho | 1 |
| Malawi | 1 |
| South Africa | 1 |
| Uganda | 1 |
| Mozambique | 1 |
| **Asia** | |
| Bangladesh | 1 |
| India | 3 |
| Iran | 1 |
| Nepal | 1 |
| Vietnam | 1 |
| **Latin America** | |
| Bolivia | 1 |
| Brazil | 1 |
| Haiti | 1 |
| **Countries classification by income levels** | |
| Upper middle-income | 2 |
| Lower middle-income | 15 |
| Low income | 8 |
| **Study design** | |
| Cross-sectional | 18 |
| Cohort | 4 |
| Intervention | 3 |
| **Outcome** | |
| Common mental disorder symptoms | 21 |
| Mental well-being | 4 |
| **Screening tools for outcome** | |
| Edinburgh postnatal depression scale (EPDS) | 1 |
| Hopkins Symptoms Checklist (HSCL-25) | 4 |
| Self-Reporting Questionnaire (SRQ-20) | 4 |
| Centre for Epidemiologic Studies Depression scale (CES-D) | 1 |
| Becks Depression and Anxiety Scale | 1 |
| WHO-5 Well-being Index | 4 |
| General Health Questionnaire (GHQ-12) | 2 |
| Composite International Diagnostic Interview (CIDI) | 1 |
| Unstandardized structured questionnaire | 5 |
| **Exposure** | |
| Water insecurity | 18 |
| Inadequate sanitation | 5 |
| Water insecurity and inadequate sanitation | 2 |

**Table 3. Methodological description of the studies included in the review.**

| Author(s) & year | Study setting | Study design | Sample size | Sampling | Exposure/measure | | Outcome/measure | |
|---|---|---|---|---|---|---|---|---|
| Aihara et al., 2016 [29] | • Urban setting of Nepal • Postnatal women admitted for delivery. | Cross-sectional | 300 | Convenience sampling | Water insecurity | • Household water insecurity scale (HWIS) • HWIS developed by adapting the household food Insecurity access scale. | Common mental disorder (CMD) symptoms | Edinburgh postnatal depression scale (EPDS) • Validated for the study setting • Cut-off point 12/13 |
| Angela et al., 2004 [22] | Two Rural communities within Paraíba state in Northeast Brazil | Cross-sectional | 204 (102 drought prone; 102 drought free) | Systematic sampling | Drought prone areas | Unstandardized structured questionnaire | • CMD symptoms | Self-reporting questionnaire (SRQ-20) • Validated for the study setting • Cut-off point of 7/8 |
| Boateng et al., 2022 [30] | Pregnant women living with or without HIV in the rural and peri-urban areas of Kenya | Cross-sectional | 183 | Quota sampling | • Water insecurity • Food insecurity | • Household water insecurity scale (HWIS-20). • Developed and validated for the study setting. • Food insecurity access scale (validated for the study setting) | CMD symptoms (depressive symptoms) | Centre for Epidemiological Studies-Depression (CES-D) scale • Validated in the similar setting |
| Brewis et al., 2019 [31] | Rural and urban communities in Haiti | Cross-sectional | 4055 | Two stage cluster sampling | • Water insecurity | Self-reported assessment (source, access, quality & quantity) | • CMD symptoms (Anxiety) | Beck's anxiety inventory • Validated for the study setting |
| | | | | | Sanitation insecurity | Crude (presence or absence of toilet) • Validated for the study setting. | • CMD symptoms (Depression) | Beck depression inventory Validated for the study setting |
| | | | | | Food insecurity | Household Food Insecurity Access Scale (HFIAS) • Validated. | | |
| Caruso et al., 2018 [26] | Women in rural communities of Odisha, India | Cross-sectional | 1347 | Stratified multistage cluster sampling | Sanitation insecurity | • 50-item sanitation insecurity measure • Validated for the study setting | Mental well being | • World Health Organization Well-being Index (WHO-5) • Validated for the setting. • Below 13 indicate poor well-being. |
| | | | | | Functional household latrine | Presence or absence of latrine | CMD symptoms | • Hopkins Symptoms Checklist (HSCL) • Validated for the setting • Cut-off point >1.75 |
| Gruebner et al., 2012 [32] | Urban slums in Dhaka Bangladesh | Cohort | 1938 | Simple random sampling | Sanitation | Presence of a toilet | Mental well-being | WHO-5 well-being index • Cut-off point below 13 indicate poor mental well-being |

*(Continued)*

**Table 3.** (Continued)

| Author(s)& year | Study setting | Study design | Sample size | Sampling | Exposure/measure | | Outcome/measure | |
|---|---|---|---|---|---|---|---|---|
| Hirve *et al.*, 2015 [33] | Women in rural community in Pune, India | Cross-sectional with a mixed method approach | 306 | Stratified random sampling | Open defecation | A structured and pre-tested survey questionnaire | CMD symptoms (psychological stress) | A structured and pre-tested survey questionnaire |
| Kangmennaang *et al.*, 2020 [34] | Urban slum dwellers in Accra Ghana | Cross-sectional using a parallel mixed method | 499 | Cluster sampling | Water insecurity | • Household water insecurity access scale (HWIAS) • Validated for the setting | CMD symptoms (Water-related emotional distress) | • Structured questionnaire with 6 questions on (worry, embarrassment or quarrels) • Validated for the setting |
| Miller *et al.*, 2021 [35] | Rural communities of Kenya | Cross-sectional | 716 | Purposive sampling | Water insecurity | • Household water insecurity access scale (HWIAS) • Validated for the setting | CMD symptoms (Probable depression) (Depression scores ≥1.75) | • Hopkins symptom checklist (HSC) • Validated for the setting • Cut-off point ≥1.75 |
| | | | | | Food insecurity | • Household food insecurity access scale • 9-item HFIAS • Validated for the setting | | |
| Khodarahimi 2014 [36] | Rural residents of Iran | Cross-sectional | 1198 | Random sampling | • Shortage of drinking water | • Coping styles with drinking water crisis scale (CSDWS) • Validated for the setting. | CMD symptoms (mental health) | • General health questionnaire (GHQ-28). • Validated for the setting. • Cut-off points >4. |
| Mushavi *et al.*, 2019 [27] | Rural residents of Mbarara Uganda | Cross-sectional | 1776 | All eligible persons included in the census | Water insecurity | • Household water insecurity access scale (HWIAS-8) • Validated for the setting. | • CMD symptoms (depression symptom severity) | • Hopkins Symptoms Checklist for Depression (HSCLD-15) • Validated for the setting. • Cut-off point >1.75 |
| Krumdiek, 2016 [37] | Pregnant women of mixed HIV status visiting health facility in rural Kenya | Cohort | 323 | All pregnant women visiting the clinic were recruited. | Water insecurity | • Household water insecurity scale • Validated in similar setting. | CMD symptoms (psychological distress) | Structured questionnaire adapted from UNICEF WASH questionnaire |
| Nicola, 2017 [38] | Women living in the Dzimauli villages in South Africa | Cohort | 300 | Simple random sampling | Water insecurity | Validated questionnaire on water insecurity | CMD symptoms (emotional distress) | Structured questionnaire • Validated |
| Shiras *et al.*, 2018 [39] | Maputo, Mozambique | Cross-sectional | 96 | Purposive sampling | Sanitation insecurity | Structured questionnaire | CMD symptoms | Structured questionnaire |
| Simiyu *et al.*, 2021 [40] | Older adults rural & urban Ghana | Cohort | 4735 | Multistage cluster sampling | Source of water | Structured questionnaire | CMD symptoms (major depression episode (MDE) | Composite International Diagnostic Interview (CIDI) • Validated for the study area • Cut-off points ≥2 and ≥4 |
| | | | | | Type of sanitation facility | Structured questionnaire | | |

(*Continued*)

**Table 3.** (Continued)

| Author(s)& year | Study setting | Study design | Sample size | Sampling | Exposure/measure | | Outcome/measure | |
|---|---|---|---|---|---|---|---|---|
| Slekiene and Mosler 2018 [23] | Rural residents in Malawi | Cross-sectional | 638 | Random-route sampling (every third household) | Safe drinking water behaviours | Structured questionnaire (Risk attitude norms abilities and self-regulations (RANAS)) | CMD symptoms (mental distress) | • Self reporting questionnaire (SRQ-20) • Validated for the study setting. • Cut-off point ≥7 |
| Stevenson et al., 2016 [24] | Women in rural communities in Ethiopia | Intervention | 347 | Cluster sampling | Water insecurity | • Household water insecurity scale 21 items • Validated for the study setting. | CMD symptoms | • Self reporting questionnaire SRQ-20 • Validated for the study setting. • Cut-off point ≥7 |
| Stevenson et al., 2012 [11] | Women in rural communities in Ethiopia | Cross-sectional | 324 | Systematic sampling | Water insecurity | Structured questionnaire • Validated for the study setting | CMD symptoms | • Falk Self Reporting Questionnaire (SRQ-F 29 item) • Validated for the study setting. • Cut-off point ≥7. |
| Thomas and Godfrey 2018 [41] | Urban residents in Ethiopia | Cross-sectional | 200 | Town divided into blocks then households randomly selected. | Water insecurity | Structured questionnaire (nine predictor variables), developed for the setting | CMD symptoms (emotional distress | • 4 variables to measure water related emotional distress • Validated in similar setting |
| Workman and Ureksoy 2017 [28] | Women in rural communities in Lesotho | Cross-sectional | 75 | Convenience sampling | Water insecurity | • Water insecurity scale 4-item • Validated for the study setting. | CMD symptoms | • Hopkins Symptoms checklist (HSCL-25) • Validated in similar setting. |
| Wutich and Ragsdale 2008 [42] | Urban residents of Bolivia | Cross-sectional | 72 | Simple random sampling | Water insecurity | Structured questionnaire | CMD symptoms (emotional distress) | • Structured questionnaire |
| Freeman et al., 2022 [43] | • Rural and peri-urban districts of Ethiopia | Interventional | 1472 | Cluster sampling | Sanitation insecurity | Standard wash indicators | Well-being | • WHO's Well-Being Index (WHO-5) |
| Ross et al., 2022 [44] | • Urban neighbourhood of Maputo Mozambique | Interventional | 424 | Cluster sampling | Sanitation insecurity | Sanitation Visual Analogue Scale (VAS) | Mental well-being | • WHO-5 mental well-being index |
| Vuong et al., 2022 [45] | Vietnam | Cross-sectional | 552 | Simple random sampling | Water insecurity | The access and behavioral indicators for water | Mental health | • Mental health composite score (MCS) • Validated for the study setting |
| Libey et al., 2022 [46] | Ethiopia | Cross-sectional | 469 | Simple random sampling | Water insecurity | Household water insecurity experience | Emotional well-being | • Structured questionnaire |

Fig 4. There was a statistically significant association between water insecurity and CMD symptoms, with an odds ratio of 5.67 (95% CI = 2.87, 11.12). The heterogeneity between studies was high ($I^2$ = 88%).

*Association between inadequate sanitation with common mental disorder (CMD).* Five studies reported continuous outcomes. The results (presented in Fig 5) indicated a statistically significant association between inadequate sanitation with common mental disorder (CMD)

**Table 4. Main findings of the included studies (n = 25).**

| Author(s) and year | Major findings |
|---|---|
| Aihara *et al.*, 2016 | • 70% of the study participants were worried about not having sufficient water.<br>• Women with higher household water insecurity scale (HWIS) scores had higher odds of having probable depression (EPDS cut off of 12/13) compared to women with lower HWIS scores (odds ratio = 1.43, 95% CI = 1.01–2.02) |
| Amanda 2019 | • 23% of the study participants reported severe water insecurity.<br>• Linear regression analysis indicated that water insecurity was predictive of mental well-being.<br>• Water insecurity was positively associated with HSCL-10 score, (moderate water insecurity, b = 0.137, p<0.001; severe water insecurity, b = 0.194, p<0.001). |
| Angela *et al.*, 2004 | • Individuals living in drought prone areas scored significantly higher scores of emotional distress compared to individuals living in drought free areas (M = 5.38 versus 3.93, F (1.200) = 5.94, p<0.01).<br>• Significant main effect of gender, with females (M = 6.21) having higher levels of emotional distress than men (M = 3.20), F (1,200) = 27, 14, p < .01. |
| Boateng *et al.*, 2022 | • Water insecurity and food insecurity interact multiplicatively to increase maternal depression.<br>• Water insecurity, food insecurity and HIV on maternal depression had synergistic interaction.<br>• 1-point increase in water insecurity was associated with 0.06-point increase in depression scores.<br>• Syndemic interaction (multiple linear regression analysis) between water insecurity, food insecurity and HIV increase depressive symptomatology (β = 0.06; p<0.05). |
| Brewis *et al.*, 2019 | • Household water insecurity exerts a significant influence on depression and anxiety through both direct and indirect pathways (food insecurity and sanitation insecurity).<br>• Water insecurity was significantly (0.01) associated with anxiety and depression.<br>• Water insecurity was significantly associated with depression by 2.66 units and anxiety scores by 2.83.<br>• The influence of WI on depression and anxiety and depression is mediated through its influence on food insecurity.<br>• Having a toilet reduced anxiety scores by 2.53 units.<br>• Women had higher anxiety and depression score compared to men. |
| Caruso *et al.*, 2018 | • 64.1% had no access to a functional household latrine.<br>• There was a positive association between a functional household latrine and mental well-being (β = 3.37, p<0.001).<br>• There was no association between a functional household latrine and anxiety, depression and distress |
| Gruebner *et al.*, 2012 | • Sanitation (presence of toilet) was found to be a factor that predicted psychological well-being.<br>• Mental well-being was positively associated with good sanitation.<br>• Sanitation significantly predicted psychological well-being (β = 0.08, p<0.001). |
| Hirve *et al.*, 2015 | • 9% practiced open defecation.<br>• There was a significant link between psychosocial stress and the practice of open defecation.<br>• Compare to latrine users a significant higher proportion (p<0.001) of open defecators reported feeling worried, irritated, rushed, tensed and depressed.<br>• Sources of stress included lack of personal safety as a woman (64%), lack of privacy (44%), and insufficient cleanliness. |
| Kangmennaang *et al.*, 2020 | • 42% of the households were water insecure.<br>• Main source (60%) of water was vended water<br>• 27% experienced emotional stress related to buying and collecting water.<br>• 21% felt angry and frustrated about the water situation.<br>• Water-insecure households were more likely to experience emotional distress compared to water-secure households (OR = 1.90, p<0.01) |
| Miller *et al.*, 2021 | • Water insecurity and food insecurity jointly were associated with lower mental health scores.<br>• Each 1-point higher water insecurity was associated with 0.64-point lower mental health summary scores, (-0.64 (95% CI = -0.94 to -0.33; p = 0.001) |

*(Continued)*

**Table 4.** (Continued)

| Author(s) and year | Major findings |
|---|---|
| Khodarahimi 2014 | • There was a significant association between drinking water shortage and mental health (M = 44.47, S. D = 13.34; p = 0.0001).<br>• GHQ-28 scores were higher (m = 46.55, SD = 3.13; p = 0.0001) among study participants affected by drought than those not affected by drought (m = 31.00, SD = 1.67; p = 0.0001). |
| Mushavi et al., 2019 | • Mean water insecurity was higher in women than men (8.7 vs 7.28) t = -2.38, p = 0.02).<br>• Positive statistically significant association between water insecurity and depression severity (b = 0.009, 95%CI (0.004 to 0.15).<br>• Estimated association was larger in men than in women, (b = 0.012, 95% CI = 0.008 to 0.015) and (b = 0.008, 95%CI = 0.004 to 0.012) respectively. |
| Krumdiek, 2016 | • 77.7% of women had at least one experience of water insecurity in the prior month.<br>• 77.3% of women had psychological stress caused by water acquisition.<br>• They felt concerned for their physical safety when accessing water. |
| Nicola, 2017 | • One third of participants expressed worry, fear, annoyance over the quality of water<br>• Felt embarrassed over their water situation.<br>• Felt anger towards inappropriate use of water.<br>• Felt ashamed because other villages had tap water which is treated.<br>• Water citizenship (the use of science and law to demand for the right of water supplied by the government) positively associated with water related emotional distress (r = 0.407, p = 0.003).<br>• Emotional distress had a significant negative correlation with education (r = -0.299, p = 0.035). |
| Shiras et al., 2018 | • Participants reported stress due to: lack of privacy, lack of safety, feeling disgust and shame, conflict with neighbours as a result of collective action failure in managing latrine.<br>• 68 of female participants mentioned concerns relation to latrine use, due to fear of physical and sexual assault. |
| Simiyu et al., 2021 | • 7.3% of the respondents had Major Depression Episode.<br>• 90% used improved water sources.<br>• 78% used improved sanitation.<br>• 77% shared sanitation facilities.<br>• Study participants who used unimproved water sources were 1.6 times likely to report Major depression episode.<br>• Individuals who used unimproved sanitation were 1.3 times likely to report MDE.<br>• Women were more significantly to be depressed than men (8.3% vs. 5.9%) p = 0.002 respectively.<br>• The effect of using unimproved water and sanitation on depression was much substantial and significant among women compared to men. |
| Slekiene and Mosler 2018 | • Significant negative relationship between mental health and safe water collection (p = 0.01, r = -0.104).<br>• More than quarter reported poor mental health.<br>• (26.8%) scored equal or above 7 on SRQ-20 (M = 4.46, SD = 3.99) |
| Stevenson et al., 2016 | • 2-point decline of water insecurity in the intervention group compared to controls (beta -1.99: 95% CI -3.15, 0.84).<br>• No direct impact of the intervention on women's psychological distress.<br>• Water insecurity was predictive of psychological distress (p<0.01, b = -2.28 (-3.77, 0.80). |
| Stevenson et al., 2012 | • Water insecurity positively associated with psychosocial distress. (r = 0.22, p<0.001<br>• Significant negative association between water quantity and psychological distress.<br>• Water insecure women experienced more symptoms of common mental disorders. |
| Thomas and Godfrey 2018 | • There was no significant association between water quantity with water related emotional distress (r = -0.09, p = 0.783).<br>• Water-related emotional distress was associated with the cost of water and the size of household. |
| Workman and Ureksoy 2017 | • Water insecurity (access, usage and perceived water cleanliness) was significantly associated with anxiety and depression. |

(Continued)

**Table 4.**  (Continued)

| Author(s) and year | Major findings |
|---|---|
| Wutich and Ragsdale 2008 | • Access to water distribution systems was significantly associated with emotional distress.<br>(r = 0.05, p = 0.03)<br>• Female gender was significantly associated with emotional distress. |
| Freeman *et al.*, 2022 | • The intervention (change behaviour related to sanitation and hygiene) had no impact on well-being, (PR: 0.90, 95%CI: 0.74, 1.10). |
| Ross *et al.*, 2022 | • Intervention respondents (used shared sanitation) experienced a 0.2 SD gain in mental well-being. |
| Vuong *et al.*, 2022 | • Using less than 50 liters of water per person per day (pppd) and the use of untreated water was significantly associated with lower mental health scores (MCS), (p<0.05). |
| Libey *et al.*, 2022 | • Higher levels of household water insecurity experiences positively correlated with-water related emotional distress, (0.57, p<0.01) |

symptoms, overall standardized mean difference (SMD) = 5.36 (95% CI = 2.51, 8.20). Heterogeneity between studies was substantial ($I^2$ = 100%).

**Sub-group analysis.** Exploratory sub-group analyses were conducted to investigate the high heterogeneity. See in figures presented in the supplementary files (S1–S3 Figs). There was no evidence of differences in the associations between water insecurity and/or inadequate sanitation and common mental disorders based on the country income category.

**Risk of publication bias.** Visual inspection of the funnel plots presented in the supplementary files (S4–S6 Figs) indicated an asymmetrical funnel plot for the continuous outcome and a symmetrical inverted funnel plot for the binary outcome, indicating possible publication bias for the continuous outcome.

**Sensitivity analysis.** Eleven high quality studies (rated as strong and moderate since most of the studies were cross-sectional studies), presented in the supplementary files (S7 Fig). There was a statistically significant association between water insecurity and common mental disorders, overall SMD = 3.68 (95% CI = 2.46, 4.89) $I^2$ = 99%, for continuous outcome, with odds ratio of 3.29 (95% CI = 1.27, 8.57) $I^2$ = 92%, for the binary outcome. Including 5 studies where confounding factors were controlled for properly (rated as strong), presented in the supplementary files(S8 Fig). There was a statistically significant association between water insecurity and CMD symptoms, overall SMD = 5.57(95% CI = 1.18,9.95) $I^2$ = 99%.

**Association between sanitation with mental well-being.** Four studies examined the association between sanitation and mental well-being. Two of these studies found a positive association between access to a functional latrine within a household compound and mental well-being. In addition, the other two studies found no association between sanitation interventions and mental well-being.

## Discussion

In this systematic review and meta-analysis, we found statistically significant associations between water insecurity and/or inadequate sanitation with CMD symptoms (as a continuous measure) and probable CMD (as binary outcome).

Water insecurity and inadequate sanitation may exert both direct and indirect effects on depression, anxiety, or stress levels. An important contributor to mental health is how individuals appraise their environment in relation to their present and anticipated living conditions. Of great importance in this appraisal is how individuals perceive suffering and harm as a result of stressors in the environment. Therefore, mental distress is considered a relative concept that mirrors the association between environmental demands, the available resources to manage

| Author & year | Selection bias | Study design | Confounding | Blinding | Data collection | Withdrawals & dropouts | Rating |
|---|---|---|---|---|---|---|---|
| Aihara 2016 | Green | Yellow | Red | Green | Yellow | Green | Yellow |
| Amanda 2019 | Green | Red | Green | Red | Yellow | Yellow | Red |
| Angela 2004 | Yellow | Red | Green | Yellow | Yellow | Yellow | Yellow |
| Boateng 2020 | Green | Red | Green | Yellow | Yellow | Yellow | Yellow |
| Brewis 2019 | Green | Yellow | Yellow | Red | Yellow | Yellow | Yellow |
| Caruso 2018 | Green | Yellow | Green | Yellow | Red | Yellow | Yellow |
| Gruebner 2012 | Green | Green | Yellow | Green | Yellow | Yellow | Green |
| Hirve 2015 | Yellow | Yellow | Red | Yellow | Yellow | Yellow | Yellow |
| Kangmennang 2020 | Green | Yellow | Yellow | Yellow | Yellow | Red | Yellow |
| Miller 2021 | Green | Yellow | Yellow | Green | Yellow | Green | Green |
| Khodarahimi 2014 | Red | Yellow | Yellow | Yellow | Red | Yellow | Red |
| Mushavi 2019 | Green | Yellow | Green | Green | Yellow | Green | Green |
| Krumdiek 2016 | Red | Green | Red | Green | Green | Red | Red |
| Nicola 2017 | Yellow | Yellow | Green | Yellow | Green | Green | Yellow |
| Simiyu 2021 | Green | Green | Green | Green | Green | Yellow | Yellow |
| Slekiene 2018 | Red | Yellow | Green | Green | Yellow | Yellow | Red |
| Stevenson 2016 | Yellow | Green | Green | Yellow | Green | Red | Yellow |
| Stevenson 2012 | Yellow | Green | Green | Yellow | Yellow | Green | Yellow |
| Thomas 2018 | Green | Yellow | Red | Red | Yellow | Green | Red |
| Workman 2017 | Green | Green | Yellow | Yellow | Yellow | Yellow | Yellow |
| Wutich 2008 | Green | Yellow | Yellow | Green | Red | Green | Green |
| Freeman 2022 | Green | Green | Green | Green | Green | Green | Green |
| Ross 2022 | Green | Green | Green | Green | Green | Yellow | Green |
| Vuong 2022 | Green | Red | Yellow | Yellow | Yellow | Green | Yellow |
| Libey 2022 | Green | Red | Yellow | Red | Yellow | Green | Red |

**Fig 2. Methodological quality assessed by EPHPP: Green = Strong, Yellow = Moderate, Red = Weak.**

these demands, and the appraisal of this association [17]. CMD symptoms as a result of water insecurity and/or inadequate sanitation manifest from the stressful experiences arising from an individual's day-to-day roles and experiences.

The findings hypothesized in qualitative studies with regards to the links between water insecurity, inadequate sanitation and mental health, are borne out by synthesis of findings from quantitative studies in this review. In this review, we found statistically significant association between water insecurity and/or inadequate sanitation and mental ill-health. This is consistent with qualitative studies conducted in sub-Saharan Africa [42, 47, 48] which found that water insecurity is linked to fear, shame, anger, worry, quarrels, and social disengagement. A study conducted in Uganda [27] found that water insecurity led to undesirable social outcomes

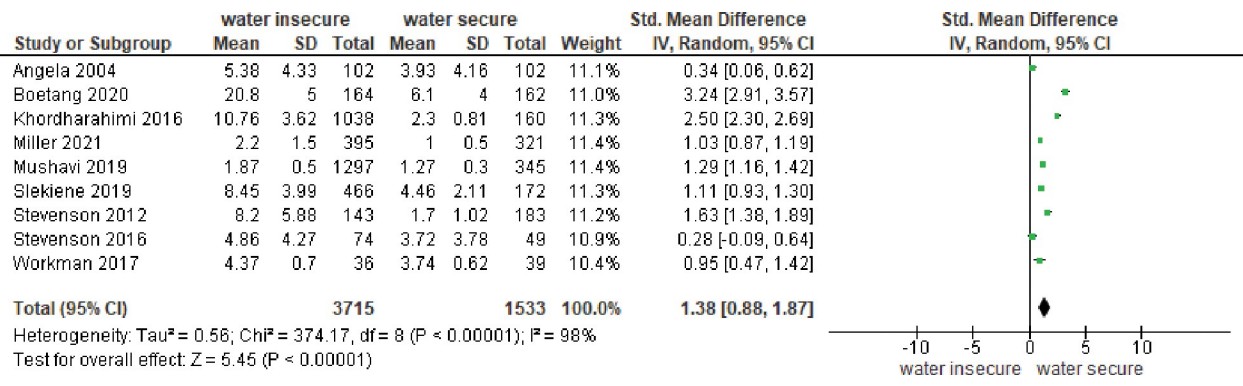

**Fig 3. Meta-analysis of association between water insecurity and CMD.**

and choice-less-ness, which led to emotional distress. Lacking water security and sanitation may compound the effects of poverty, leading to mental health problems due to shame and failure to fulfil or meet social expectations, frustrations due to the time lost when fetching or queuing, interpersonal conflicts and perceived unjust treatment.

A qualitative synthesis [49] that explored the relationship between sanitation and mental well-being, found that people experience lack of privacy and safety when using sanitation facilities or during open defecation, and that this influenced their mental and social well-being. In addition, other studies [50, 51], found that women sometimes withhold water and food to limit urination, suppressing their needs due to the poor physical and social environment, felt helpless about sanitation situation, and feared or experienced physical or sexual abuse while accessing defecation sites.

People with low socio-economic status are more likely to live in neighbourhoods with water insecurity and inadequate sanitation. In addition, poverty may affect household income levels, influencing access to water and sanitation coping strategies which in turn can affect mental health and well-being [52].

Poverty can be both a determinant and a consequence of poor mental health [53]. People with mental ill-health have increased risk of drifting into, or remaining in, poverty and this affects their access to basic needs such as household water and sanitation.

A synergistic effect of water insecurity on mental distress has been observed when the household also has food insecurity [30]. Syndemics involves a cluster of two or more diseases or health conditions in a community, in which there is some level of detrimental environment that aggravate deleterious health effects of some or all diseases involved [54]. Syndemic theory

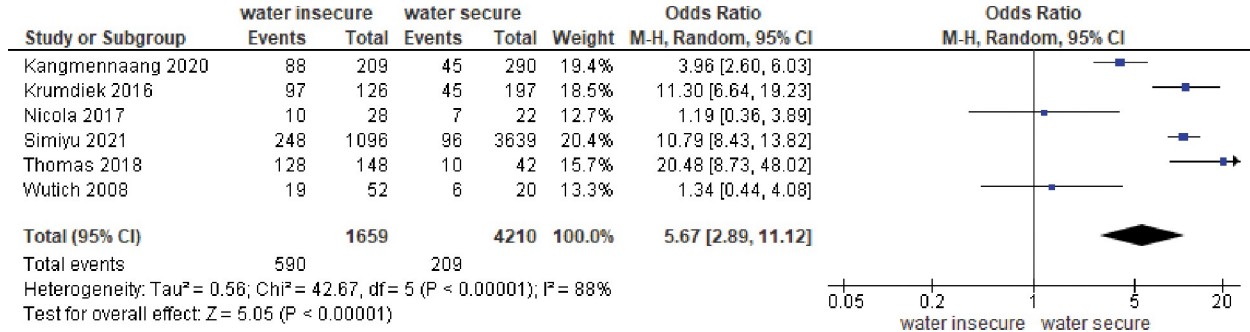

**Fig 4. Meta-analysis of association between water insecurity and CMD.**

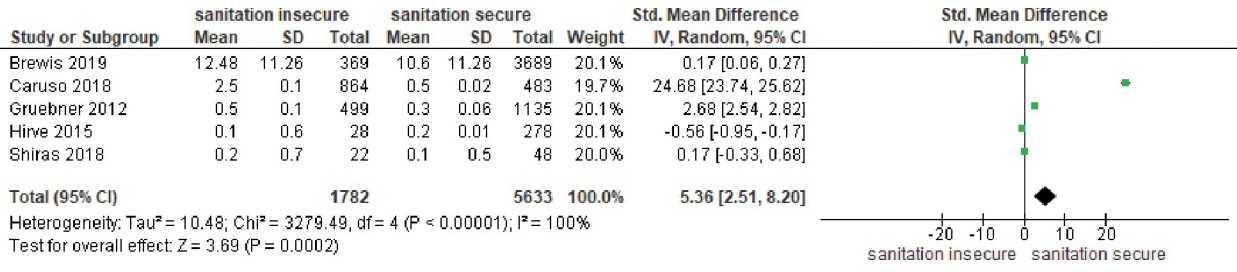

**Fig 5. Meta-analysis association between sanitation insecurity with CMD.**

offers an effective framework for explaining the complex associations between resource insecurities and mental health conditions. It is useful in understanding mechanisms through which complex relationships and interactions of factors occur in certain groups of people. Syndemic interaction between water insecurity and food insecurity may increase the effect on mental distress. In addition, other syndemically occurring health conditions include infections such as reproductive tract infections (RTI), urinary tract infections (UTI), gender-based violence and depression.

A review [55] on food insecurity and mental health from ten different countries, found a positive relationship between food insecurity and the risk of depression and anxiety. Water insecurity and food insecurity may exacerbate or worsen mental distress. Water insecurity and food insecurity are distinct, although related forms of resource insecurities [56]. Water insecurity is a fundamental driver of food insecurity [57]. Household water insecurity have a strong direct independent effect on depression and anxiety levels, even once household food insecurity is considered. Additionally, household water insecurity have an indirect effect on depression and anxiety through its influence on household food insecurity [31].

One study [24] included in this review was conducted in Ethiopia, to assess household water insecurity and women's psychological distress before and after water access improvements. The authors found that while improvement of water supply reduced household water insecurity, it was not effective in alleviating women's psychological distress. A syndemic approach may help in understanding complex relationships and interactions between water insecurity and mental ill-health. This will help in finding out other factors that are associated with water insecurity and mental ill-health, and in coming up with effective interventions.

## Strengths and limitations

Strengths for this study included the rigorous approach to screening and data extraction, and use of meta-analysis. Limitation in the studies included in the review was that the majority (18 out of 25) of the included studies were cross-sectional studies. They therefore failed to identify and detect changes over time and to provide insights into causal relationships between water and sanitation insecurity with mental distress. Furthermore, in cross-sectional studies, a more negative appraisal of environmental conditions may be driven by negative recall bias in those with poor mental health. Effective public health practice project (EPHPP) tool was used for quality assessment for the included studies. This tool fails to evaluate whether studies were adjusted for clustering, this may underestimate standard error and confidence interval. We limited our search to English language, and we did not search from grey literature, perhaps there could be relevant grey literature in this area. Another limitation in this review was substantial heterogeneity within studies. Sources of heterogeneity were explored using sub-group analysis using pre-specified sub-groups to find out the sources of heterogeneity; this offered little. Heterogeneity in this review could be related to the variation in the population and study design.

### Future directions

To better understand the association between water and sanitation insecurity with mental health, future longitudinal studies should be conducted. Other types of statistical analysis, such as structural equation modelling and path analysis, should be used to understand the causal relationship between water and sanitation insecurity and mental health. This would make a valuable contribution to research on water and sanitation insecurity and mental health globally and provide robust data for policymakers to make effective and practical decisions. Interventions to provide basic water and sanitation, especially for women, could substantially contribute to reducing the burden of CMD among other health and social benefits in LMICs [58]. Interventional studies will provide an understanding on the relative impact of water supply intervention on water insecurity and justify water security as an exposure, when it is really an intermediary outcome of water supply. In addition, psychosocial interventions are also needed to improve mental health.

## Conclusions

In conclusion, this systematic review and meta-analysis suggested that water and/or sanitation insecurity contributes to poorer mental health globally, independent of other poverty indicators. Most of the studies included in this review were measuring associations and have a weak claim on causal inferences. Longitudinal studies with longer follow-up times are warranted to evaluate the possible cause-and-effect relationship between water insecurity, inadequate sanitation and mental health.

## Supporting information

**S1 Checklist.**
(DOCX)

**S1 Appendix. Full search strategy.**
(DOCX)

**S2 Appendix. Effective Public Health Practice Project (EPHPP) quality assessment tool.**
(DOCX)

**S3 Appendix. Data set for meta-analysis.**
(XLSX)

**S1 Fig. Subgroup analysis on the associations between water insecurity and common mental disorders based on the country income category.**
(TIF)

**S2 Fig. Subgroup analysis on the association between water insecurity and common mental disorders based on the country income category.**
(TIF)

**S3 Fig. Subgroup analysis on the association between inadequate sanitation and common mental disorders based on the country income category.**
(TIF)

**S4 Fig. Funnel plot for continuous outcome.**
(TIF)

**S5 Fig. Funnel plot for continuous outcome.**
(TIF)

**S6 Fig. Funnel plot for binary outcome.**
(TIF)

**S7 Fig. Sensitivity analysis for high quality studies.**
(TIF)

**S8 Fig. Sensitivity analysis for studies where confounding factors were controlled for properly (rated as strong).**
(TIF)

## Author Contributions

**Conceptualization:** Joan J. Kimutai, Charlotte Hanlon.

**Data curation:** Seble Shewangizaw.

**Methodology:** Joan J. Kimutai, Merga Feyasa.

**Software:** Joan J. Kimutai, Merga Feyasa.

**Supervision:** Crick Lund, Wilkister N. Moturi, Charlotte Hanlon.

**Writing – original draft:** Joan J. Kimutai.

**Writing – review & editing:** Crick Lund, Wilkister N. Moturi, Charlotte Hanlon.

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
