## [Decision Letter · Decision Letter 0]

1 Mar 2023

PONE-D-23-00344Evidence on the links between water insecurity, inadequate sanitation and mental health: a systematic review and meta-analysisPLOS ONE

Dear Dr. Kimutai,

Thank you for submitting your manuscript to PLOS ONE. After careful consideration, we feel that it has merit but does not fully meet PLOS ONE’s publication criteria as it currently stands. Therefore, we invite you to submit a revised version of the manuscript that addresses the points raised during the review process. All three reviewers are very complimentary about the paper but suggest some things to address including extra papers to include.

We look forward to receiving your revised manuscript.

Kind regards,

Alison Parker

Academic Editor

PLOS ONE

Journal Requirements:

“This work was supported by the Centre for Innovative Drug Development and Therapeutic Trials for Africa (CDT-Africa), Addis Ababa University.”

“The authors received no specific funding for this work.”

4. Please ensure that you refer to Figure 3 in your text as, if accepted, production will need this reference to link the reader to the figure.

5. We note you have included a table to which you do not refer in the text of your manuscript. Please ensure that you refer to Table 2, 3 and 4 in your text; if accepted, production will need this reference to link the reader to the Table.

Reviewers' comments:

Reviewer's Responses to Questions

**Comments to the Author**

1. Is the manuscript technically sound, and do the data support the conclusions?

Reviewer #1: Yes

Reviewer #2: Partly

Reviewer #3: Yes

2. Has the statistical analysis been performed appropriately and rigorously? 

Reviewer #1: Yes

Reviewer #2: Yes

Reviewer #3: Yes

3. Have the authors made all data underlying the findings in their manuscript fully available?

Reviewer #1: Yes

Reviewer #2: No

Reviewer #3: Yes

4. Is the manuscript presented in an intelligible fashion and written in standard English?

Reviewer #1: Yes

Reviewer #2: Yes

Reviewer #3: Yes

5. Review Comments to the Author

Reviewer #1: I think this is an important paper and one that needs to be published. I have only a couple areas that I feel need to be improved.

1.) The figures are very pixilated (low resolution) and difficult to read. It would be helpful if these could be redone in high resolution.

2.) Since the database search July 2022 till now (Feb 2023), there have been a few more papers published on this topic including at least one authored by Ian Ross (https://scholar.google.com/citations?user=gHAjfvcAAAAJ&hl=en). I know that a date must be chosen at some point but it would seem a good idea to update the search maybe up to the beginning of 2023.

Reviewer #2: This is a reasonably useful review on an important topic. It does have some important weaknesses, but they can be fixed. Some of the remedies I suggest may be unappealing, but would turn an OK paper into a great paper that will attract 5x more cites.

Six major comments:

1. Search strategy omits water supply

The authors have been fairly rigorous on their approach to the outcome (mental health) but unfortunately a bit weak on the exposures. The authors imply at the bottom of p.5 that they are building on Bisung and Elliot’s 2017 scoping review to make it systematic. However, for water terms, the authors search only for “(water insecurit* OR water scarcit*)”. I think the authors need to clarify how they are defining “water insecurity”, and whether inadequate access to water supply (using their sanitation terminology) is part of that, or only self-perceived water security (like HWISE and similar). Without this definition, it is unclear whether an ostensibly water supply oriented paper would be included or not. For example, if a piped water intervention study included mental wellbeing as an outcome, should that be included in this review even if a psychometric measure of water insecurity was not included (i.e. the exposure comparison would be two levels of water *service*).

Bisung & Elliot’s water search terms were “‘water supply’ or ‘water *security’ or ‘water access’ or ‘access to water’ or ‘water distribution*’ or ‘water availabilit*’ or ‘water quan*” which represent a more rounded and comprehensive search strategy. By focusing on only two search terms, and apparently a narrow definition of water insecurity, the authors have almost certainly missed relevant literature. The authors have mostly treated sanitation as a service (e.g. inadequate/poor sanitation) but not treated water in conceptually the same way. Their sanitation terms are “(inadequate sanitation OR sanitation insecurit* OR poor sanitation OR open defecation)”. Even then, these sanitation terms are far from ideal – why not just “sanitation”? this has undoubtedly resulted in excluded studies (see below).

Recommendation: authors re-run searches including more appropriate search terms for water supply and sanitation, de-duplicate, and include any additional studies identified. I understand that this may be unappealing. An alternative could be to keep the same results as now, but (i) justify in methods why water supply terms were not included; (ii) acknowledge the lack of water supply terms in the discussion as a substantial limitation; (iii) clearly define water security and why inadequate water supply is not part of it. I leave it to the editor to decide what to insist on. If the authors re-run searches they could also include the “missing” papers I note below. The authors might note that they only got 1599 EMBASE hits for their specific water search terms, but an order of magnitude for having used then “/water supply” subject header, a redeeming feature of an apparently flawed search strategy. But why include it is a subject term if not as a search term?

2. Missing papers

This paper (https://journals.plos.org/globalpublichealth/article?id=10.1371/journal.pgph.0000056)

was published in January ’22 and the searches were to July ‘22, so should have been picked up since it meets the inclusion criteria. It is unclear why it was not. This is evidence of the weak search strategy. I think it’s the first sanitation intervention study to include mental wellbeing outcomes.

This paper was published in October ’22 so after the searches, but the authors should mention it in discussion as (I think?) the only other example of an evaluation of a sanitation intervention which measured a mental wellbeing outcome.

https://bmjopen.bmj.com/content/12/10/e062517

I don’t know the water supply/security literature as well but my impression is that, given the weak search strategy, and the fact that the Freeman paper was missed, probably relevant water papers have been missed as well.

Here’s a water one from Oct ’22 https://www.sciencedirect.com/science/article/abs/pii/S1438463922001420

This supports the idea of clarifying the definitions and search strategy, re-running the searches, and updating the paper. I expect 5-10 additional papers would be included.

3. Engaging with what studies controlled for

Most meta-analyses focus on intervention studies. That way, depending on the quality of causal inference, you can be reasonably confident that the intervention caused a change in the outcome. All but one of the studies are not intervention studies. That makes it doubly-important to engage with what the studies controlled for. Otherwise everything is likely confounded with poverty in one way or another. Poorer people are more likely to be anxious/depressed etc., and poor people are more likely to be water insecure. Hey presto, water insecurity and depression are correlated! But it’s only if you adjust for poverty/wealth etc. that you’ll see the true effect size. This issue was not properly engaged with in this paper. The authors have “confounding” as an item in study quality (fig 2) but it’s not clear at all how this is defined/measured – see below on this. Table 3 or Table 4 really needs to engage with the confounding/adjustment issue. And I would also recommend conducting a sensitivity analysis excluding studies which did not control for confounders properly. Overall I think the pooled effect sizes are likely to be overestimated for this reason.

4. lack of relevant sensitivity analyses

Only looking at SGA by income group is not enough. Study quality is evaluated but the results are not used. The results of that exercise should be used to calculate an overall score per study, then include a sensitivity analysis in which only studies above X score (or excluding the bottom 2 quintiles for quality) are included, to see if results are being biased by a few low-quality studies. Otherwise what’s the point of reviewing study quality? Likewise ideally a SA on studies that did/didn’t adjust for clustering properly.

5. Study quality appraisal

An annex is needed which explains how each item in the EPHPP tool was defined. Were these the Qs used? https://www.ephpp.ca/PDF/Quality%20Assessment%20Tool_2010_2.pdf Just add the Qs to an annex, so it’s transparent, alongside how answers were converted into the red/yellow/green in figure 2. This is basic replication stuff. All systematic reviews should be replicable. As far as I can tell, this EPHPP study quality tool doesn’t evaluate whether studies adjusted for clustering. Studies which don’t do this, and there are many in WASH (Wolf 2022 explain - https://www.thelancet.com/journals/lancet/article/PIIS0140-6736(22)00937-0/fulltext) will have underestimated standard errors and therefore 95% CIs. The authors need to flag this as a limitation.

6. Open science

Authors should put their meta-analysis datasets and analytical code underlying their results on a public repository like OSF.io. This should just be a given nowadays. Everything should be replicable, and what’s more it is PLOS policy - https://journals.plos.org/plosone/s/data-availability . The authors state “All relevant data are within the manuscript and its supporting information files” but that is not really true. Publish the actual dataset you analyse for the meta-analysis.

Other points

Abstract

- recommend the authors review PRISMA 2020 for abstracts, because their abstract is missing many several of the items. The PRISMA website is down, but here it is elsewhere https://journals.plos.org/plosntds/article/file?type=supplementary&id=10.1371/journal.pntd.0010822.s002

Introduction and methods

- p5 line L13 – also re sanitation there is disgust at seeing/smelling faeces which can trigger anxiety/shame. https://www.sciencedirect.com/science/article/pii/S0277953621000411

- p.6 3 lines from bottom (please use line numbers!)– clarify to “mental well-being” if that is what was searched for because wellbeing is a far broader concept. You haven’t included studies of water security and happiness for example. https://link.springer.com/article/10.1007/s10902-018-0060-6

- I’m assuming the DerSimonian and Laird random effects model was used. Best just to state that as there are alternatives which are more and more popular.

Results

- Shiras et al 2018 is really a qualitative study as it’s based on semi-structured IDIs – it should probably be excluded. But it’s up to the authors if they think those IDIs are providing quant data, because it is borderline.

- Table 4 on the whole is v useful. However, it needs to note p-values (ideally, but 95% CI OK as well) at all times when an association is claimed. It also needs to explain how water insecurity (or whatever the exposure is) was defined in each study. This should be in a supp mat table if no space. I would wager they are sufficiently different that the meta-analyses are comparing apples and pears, but that’s a discussion for another day.

- For the Stevenson 2016 study, the authors say “We did not find evidence of impact of the intervention on women’s psychological distress. Water insecurity was, however, predictive of psychological distress (p <0.01), independent of household food security and the quality of the previous year’s harvest.” So table 4 is slightly misleading re: this being an intervention study. The authors are using the observational result rather than the intervention result (which was no effect). And this is the ONLY study which was an intervention study! Needs to be clarified in table 4 and discussion (we need more intervention studies which measure these outcomes).

- For the SMD results and meta-analysis, it’s not clear what is being compared (difference between what and what?). this could be addressed in either table 3 or table 4 (or in a supp mat table)

Discussion

- Needs to more clearly state that there are no intervention studies! In order to allocate resources, we need to understand the relative impact of water interventions on water security. This paper goes into how using HWISE or similar can help inform resource allocation https://www.sciencedirect.com/science/article/pii/S0043135422012726

- Talk about water supply interventions, need to justify focus on water security as an exposure, when really it’s an intermediary outcome of water supply. the food security points are not that relevant. Need more attention on the above points and the limitations.

- the conclusion needs to be clearer that most of these studies are only measuring associations and have a weak claim on causal inference. Cf. issues with poverty confounding above. Suggest “we provide some evidence of an association between X and Y”

Reviewer #3: This is a well-written and important contribution to the literature surrounding water/sanitation access and health indicators. The analysis is thorough and careful, and the authors are indeed filling a gap in the academic literature. In my opinion, this paper will be a welcome contribution to PLOS ONE with the following suggested edits:

- On page 10, the authors note "Review Manager (RevMan) software was used for the meta-analysis. A random effects statistical model for meta-analysis was used..". The methodology should be explained in more detail here - explaining how the software gleans insight from the information provided to it. More detail would include why a random effects model was used.

- On the same page, more detail should be provided on how a forest plot provides insight into publication bias. In other words, the methodology needs to be laid out for the reader.

- Figures 1 and 2 have fonts that are difficult to read - they may need to be re-created. Figures 9-11 require that the axes labels be spelled out (rather than the use of acronyms only).

6. PLOS authors have the option to publish the peer review history of their article (what does this mean?). If published, this will include your full peer review and any attached files.

Reviewer #1: No

Reviewer #2: No

Reviewer #3: No

---

## [Author Response · Author response to Decision Letter 0]

5 Apr 2023

Response to Academic Editor

-Our manuscript meets PLOS ONE’S style. We used The PLOS ONE style template.

-We used “Fig1.tif” style to name all our figures and “Table 1” for all the tables in the manuscript.

2. Acknowledgments Section

-We removed funding information in the manuscript.

-Funding statement reads as follow in the cover letter, “The authors received no specific funding for this work”.

3. Data Availability statement

-We included excel sheet of the dataset used in meta-analysis in the supplementary files.

4. Please ensure that you refer to Figure 3 in your text as, if accepted, production will need this reference to link the reader to the figure.

-We referred figure 3 in the text.

5. We note you have included a table to which you do not refer in the text of your manuscript. Please ensure that you refer to Table 2, 3 and 4 in your text; if accepted, production will need this reference to link the reader to the Table.

-We referred table 2,3 and 4 in the text.

Guidelines for resubmitting your figure files.

-We used PACE to convert the tables and figures to a resolution that is more visible.

Response to Reviewers

Response to reviewer 1

1.) The figures are very pixilated (low resolution) and difficult to read. It would be helpful if these could be redone in high resolution.

We used Preflight Analysis and Conversion Engine (PACE) digital diagnostic tool, and now the figures are in high resolution.

2.) Since the database search July 2022 till now (Feb 2023), there have been a few more papers published on this topic including at least one authored by Ian Ross (https://scholar.google.com/citations?user=gHAjfvcAAAAJ&hl=en). I know that a date must be chosen at some point but it would seem a good idea to update the search maybe up to the beginning of 2023.

We updated our search to March 2023, and in our review, we added the study authored by Ian ross with colleagues.

Response to Reviewer 2

1. Search strategy omits water supply.

We revised our search strategy to represent a more rounded and comprehensive search strategy (Pg. 8). 

‘Water *insecurity’ OR ‘water scarcity’ OR ‘access to water’ OR ‘water access’ OR ‘water availability’ OR ‘water supply’ OR ‘water quan*’ OR ‘water distribution’ OR ‘sanitation’ OR ‘open defecation’ OR ‘toilet facility*’ OR ‘latrine’ OR ‘poor sanitation’ AND ‘mental health’ OR ‘mental disorders’ OR ‘common mental disorders’ OR ‘depression’ OR ‘anxiety’ OR ‘mental distress’ OR ‘well-being’ OR ‘wellbeing’.

We did a re-run search for the articles, and we got 6105 non-duplicates articles. From Embase 4183, psych info 554, and PubMed 1368.

 In total we included 25 quantitative articles in the review.

2. Missing papers

-We updated our search to March 2023.

-Added the paper by Freeman et al., 2022, in the included studies.

-Added a paper by Ross et al., 2022, in the included studies.

-Added a paper by Vuong et al., 2022, in the included studies.

- Added a paper by Libey et al., 2022, in the included studies.

3. Engaging with what studies controlled for

Most of the included studies to conduct meta-analysis in this review were cross-sectional studies which failed to identify and detect changes over time and to provide insights into causal relationships between water and sanitation insecurity with mental distress. In this case we included it as one of the limitations of the included studies.

We did sensitivity analysis for the studies where confounding factors were controlled for properly. We found a statistically significant association between water insecurity and common mental disorders. Confounding is defined in the Effective Public Health Practice Project (EPHPP) too. Confounding is a distortion of the association between the exposure and the outcome which occurs when the study groups differ and influence the results. Here we identified studies where there was an important difference between the study participants in the study.

4.lack of relevant sensitivity analyses

-We conducted sensitivity analysis for the high-quality studies (strong and moderate). We found a statistically significant association between water insecurity and common mental disorders. We included those studies that were rated as moderate since most of the included studies were cross-sectional studies.

5. Study quality appraisal

-Presented the effective public health practice project (EPHPP) tool in the supplementary files. In addition, we explained how the answers were converted to green, yellow and red in the quality assessment diagram. In addition, in the limitation section we added the use of the EPHPP tool as a limitation. This tool fails to evaluate whether studies were adjusted for clustering, this may underestimate standard error and confidence interval.

Other points

Abstract

Added the risk of bias and synthesis of results in the abstract. 

Introduction and methods

-Added literature on how seeing and smelling faeces may trigger anxiety or shame.

-Specified the type of random effect model to - Random effect (Der Simonian and Laird) statistical model for meta-analysis

Results

-Shiras et al 2018 study provided quantitative results so we included it in the review.

Discussion

Included the following statements in the discussion:

-Interventional studies will provide an understanding on the relative impact of water supply intervention on water insecurity and justify water security as an exposure, when it is really an intermediary outcome of water supply.

Limitation

Effective public health practice project (EPHPP) tool was used for quality assessment for the included studies. This tool fails to evaluate whether studies were adjusted for clustering, this may underestimate standard error and confidence interval.

Conclusion

Included the following statements in the conclusion section.

-Most of the studies included in this review were measuring associations and have a weak claim on causal inferences. 

Response to reviewer 3

1. On page 10, the authors note "Review Manager (RevMan) software was used for the meta-analysis. A random effects statistical model for meta-analysis was used..". The methodology should be explained in more detail here - explaining how the software gleans insight from the information provided to it. More detail would include why a random effects model was used.

-The review manager software conducts meta-analysis by calculating the standardized mean difference (the sum of the weighted effect sizes, divided by the sum of weightings) when the outcome of research is continuous, and odds ratio when the outcome of research is binary. 

-Random effects model was used because it assumes the true effect may vary from study to study due to heterogeneity (differences) among studies. The effect estimates of all studies are assumed to be drawn from a normal distribution. The pooled estimate is the average or mean effect. The effect sizes in the studies used are assumed to represent a random sample for all possible effect sizes.

2. On the same page, more detail should be provided on how a forest plot provides insight into publication bias. In other words, the methodology needs to be laid out for the reader.

- A forest plot is a graphical display of findings of meta-analysis. It summarizes the data of individual studies and gives a visual suggestion of the amount of heterogeneity and the estimated common effect.

-A funnel plot is a graphical representation of the relation between the study’s effect size and its precision (effect estimated from individual studies against sample size). An asymmetry funnel plot indicates a possibility of publication bias. Whereas a symmetrical inverted funnel plot indicates no publication bias in the included studies.

3. Figures 1 and 2 have fonts that are difficult to read - they may need to be re-created. Figures 9-11 require that the axes labels be spelled out (rather than the use of acronyms only).

-We used Preflight Analysis and Conversion Engine (PACE) digital diagnostic tool, and now all the figures are in high resolution.

-For figures 9-11 I added the axes spelled out.

---

## [Decision Letter · Decision Letter 1]

10 May 2023

Evidence on the links between water insecurity, inadequate sanitation and mental health: a systematic review and meta-analysis

PONE-D-23-00344R1

Dear Dr. Kimutai,

We’re pleased to inform you that your manuscript has been judged scientifically suitable for publication and will be formally accepted for publication once it meets all outstanding technical requirements.

Kind regards,

Alison Parker

Academic Editor

PLOS ONE

Additional Editor Comments (optional):

Reviewers' comments:

Reviewer's Responses to Questions

**Comments to the Author**

1. If the authors have adequately addressed your comments raised in a previous round of review and you feel that this manuscript is now acceptable for publication, you may indicate that here to bypass the “Comments to the Author” section, enter your conflict of interest statement in the “Confidential to Editor” section, and submit your "Accept" recommendation.

Reviewer #1: All comments have been addressed

2. Is the manuscript technically sound, and do the data support the conclusions?

Reviewer #1: Yes

3. Has the statistical analysis been performed appropriately and rigorously? 

Reviewer #1: Yes

4. Have the authors made all data underlying the findings in their manuscript fully available?

Reviewer #1: Yes

5. Is the manuscript presented in an intelligible fashion and written in standard English?

Reviewer #1: Yes

6. Review Comments to the Author

Reviewer #1: In my opinion the authors have adequately responded to the reviewer comments. I believe this manuscript is ready for publication.

7. PLOS authors have the option to publish the peer review history of their article (what does this mean?). If published, this will include your full peer review and any attached files.

Reviewer #1: No

---

## [Editor Report · Acceptance letter]

17 May 2023

PONE-D-23-00344R1 

Evidence on the links between water insecurity, inadequate sanitation and mental health: a systematic review and meta-analysis 

Dear Dr. Kimutai:

I'm pleased to inform you that your manuscript has been deemed suitable for publication in PLOS ONE. Congratulations! Your manuscript is now with our production department. 

Kind regards, 

on behalf of

Dr. Alison Parker 

Academic Editor

PLOS ONE